# Location problem of *Osmia cornuta* nesting aids for optimum pollination

Juraj Pekár◉, Marian Reiff◉*, Ivan Brezina◉

Faculty of Economic Informatics, Department of Operations Research and Econometrics, University of Economics in Bratislava, Bratislava, Slovakia

◉ These authors contributed equally to this work.
* marian.reiff@euba.sk

## Abstract

The economic contribution of insect pollinators is evident as they contribute to higher crop yield quantity and quality. The management of bee species is key to crop production, especially where wild and domesticated bees are in low abundance. Several bee species have been identified as possible candidates for replacing, or at least supplementing, the decreasing number of honey bees. Our research seeks to address the location problem as regards nesting aids for *Osmia cornuta* bees in orchards using mathematical programming models for determining the optimal location of nesting aids and optimizing the management of solitary bees. A differential evolution algorithm is used to solve a location model of *Osmia cornuta* nesting aids for optimum pollination. Instead of a random ad hoc location of nesting aids in an orchard, or at the edge of an orchard, we utilize an effective optimization tool to determine locations which will optimize pollination by alternative pollinators, and increase the economic output of an agricultural business. The importance of this proposed model is likely to increase with agricultural intensification, and the decrease of the numbers of wild pollinators.

## 1. Introduction

Insect pollination is a valuable ecosystem service for agriculture crops. While wind-pollination and self-pollination occurs for some crops, insect pollination generally increases yields and the quality of produce. Many field crops require an operating pollination system which influences the productivity of approximately 75% crop species [1], and 80% of all flowering plant species rely on animal pollinators [2]. The acreage of insect pollination crops has grown substantially in recent decades, resulting in higher demand for pollination services [3]. At the same time, a worldwide decline in the abundance and diversity of pollinators has been recorded in recent years [4–6]. Agricultural intensification lowers pollinator abundance and diversity and limits wild bees to resource islands in an unrewarding matrix [7–9]. Pollination is an economically important ecosystem service that is threatened by biodiversity losses [10] and insufficient pollination is a common cause of low yields, such as [11,12].

**Data Availability Statement:** All relevant data are within the manuscript and its Supporting information files.

**Funding:** The Grant Agency of the Slovak Republic supported this work – VEGA grant no. 1/0339/20

https://www.minedu.sk/about-the-ministry/ The funders had no role in study design, data collection and analysis, decision to publish, or preparation of the manuscript.

**Competing interests:** The authors have declared that no competing interests exist.

Although honey bees (*Apis mellifera*) can be used as pollinators in large commercial orchards to improve productivity [13], studies have shown several other insects, notably solitary bees (e.g., *Osmia spp. Andrrena spp.*) and bumblebees (*Bombus spp.*) to be effective alternative pollinators of, for example, apple, cherry, pear, and strawberry crops and in some cases more effective than honeybees [14]. The advantages of alternative pollinators include the lower temperature range when pollinators are active, their preference for foraging and, stigma contact by floral visitors.

For example, temperature and relative humidity has been shown not to effect *Osmia cornuta* presence in orchards [15,16]. Commercially high yields in *Osmia* pollinated orchards have been achieved, even where there was poor weather during bloom [17,18].

A promising pollinator species would prefer to forage on flowers of the target crop [19]. For example, *O. cornuta* bees do not seek out other food sources in the event of a pollen decline in a pear orchard, in contrast to honeybees [20]. In addition, pollination influences the number of seeds, and seed sets are an important factor in pear commercialization. Parthenocarpic fruits are often smaller than seeded ones, and size is a crucial factor as regards customer preference. Pears with a high numbers of seeds tend to be larger and have better shape and flavor [21]. Positive effects of the pollinator *O. cornuta* on strawberry quality, namely size, weight, shape, and color, have been demonstrated by a study [22] and the pollinator *Osmia bicornis* has been shown to positively affect pollination and fruit quality in commercial sweet cherry orchards by a study [23]. High rates of stigma contact by floral visitors are strongly related to the pollinator's behavior on flowers. It is known that honey bee behavior results in low rates of stigma contact, whereas pollen collectors (e.g., *Osmia*) are much more efficient pollinators of crops [24–26]. Rates of stigma contact are sometimes also related to the pollinator's body size relative to the crop flower's size. In addition, in orchards where non- *Apis* bees were present, the foraging behavior of honey bees changed, and the pollination effectiveness of a single bee visit was more significant than in orchards where non- *Apis* bees where absent. This difference translated to a greater proportion of fruit sets in these orchards [27].

The contribution of insect pollinators to economic output is evident [28]. Several alternative bee species have been identified as able to replace or supplement the decreasing number of honey bees [29,30]. A literature review of establishing and managing bee species as crop pollinators can be found, for example in [31,32]. Our research looks at the location of nesting aids for solitary bees in orchards and uses mathematical programming models to determine the optimal location of nesting aids to optimize the management of solitary bees in orchards. The current practice with honey bees is as follows. Beehives are usually kept on a trailer located at the edge of a field or orchard, where the plants to be pollinated are located. This location is convenient for the beekeeper, but some distant parts of the orchard may not be covered, and may not be pollinated, as the bees need to fly long distances, or forage on another closer crop. For solitary bees, the problem may be more evident, as they fly smaller distances than honey bees. In the next part of the paper, a literature overview of mathematical programming location models is provided. A mathematical programming model for determining the optimal location of nesting aids is proposed and applied to different case study problems to illustrate the model practical use. A differential evolution algorithm is used to solve a location model of *Osmia cornuta* nesting aids for optimum pollination.

Deciding on the location of objects is a problem that many companies face in practice. The importance of this issue, known as the location problem, has received much attention in scientific journals. Location models are among the best known and most used optimization models in practice. Location science, the basis of the object location problem, can be found as early as the 17th century, when the French mathematician Pierre de Fermat, a pioneer of analytical geometry, formulated the problem of finding a point in a plane that the sum of Euclid

distances from the three points entered is minimal. The general formulation of the geometric median is currently known as the Weber problem [33]. It requires finding a point in a plane that minimizes the sum of the transportation costs from this point to *n* destination points, where different destination points are associated with different costs per unit distance. In general, location science can be characterized as a scientific area involving many problems and a range of knowledge, both in theoretical terms and modeling approaches and solution techniques. Many classic location models are based on finding a compromise between proximity (distance) and cost. In general, the distance is a distance between demand and service objects and represents the quality of the servicing system's design. A comprehensive overview of the classification of location models is given in [34].

In the literature, the two most frequently used access distance concepts are demand-weighted distance and coverage distance. Demand-weighted distance is based on the weighted average distances between individual destinations and their assigned service objects. An example is a firm's supply chain for moving their goods or services to the consumer. A representative model of this concept is the *p*-median problem, formulated independently by Hakimi [35,36] and ReVelle and Swain [37]. The *p*-median problem seeks the location of a given number of service facilities to minimize the total demand-weighted distance between destination facilities serviced by a given number of *p* service facilities. The *p*-median problem is derived from the original Weber problem and can be found, for example, in [35–39], etc. An example of an application from pollination services, i.e. hive allocation utilizing the *p*-median problem can be found, for example, in [40].

The coverage distance concept is based on the selection of an optimal set of destination facilities at minimum cost, where all destination facilities must be served (covered) by at least one of the service facilities and all destination facilities are covered at the prescribed coverage distance from the service facilities [41]. The set covering problem is one of the basic models of location science [42,43]. The aim is to assign a destination point to a service object within a radius specified by the maximum distance, in order to locate and operate the minimum number of service objects and to cover all the destination points. In general, there are two basic modifications to the set covering problem. The *p*-centers problem is a case where the coverage distance requirement is relaxed, and for maximum covering problems, where the total coverage requirement is relaxed [44]. The *p*-centers problem is a minimax problem. It seeks the location of *p* facilities. Each demand point receives its service from the nearest facility. The objective is to minimize the maximal distance for any demand point to the nearest facility [45]. This problem has many modifications, the most common include: the capacitated *p*-center problem [46,47], the conditional *p*-center problem [48,49] and the continuous *p*-center problem, [50,51]. Maximal covering problems are models for locating service facilities using coverage criteria that take into consideration that the service facilities do not cooperate. The optimization criterion assumes that a set of non-overlapping circles (representing the location of service facilities) is sought to maximize their common radius [52].

In most of the models we have looked at so far, the number of service facilities to be located is an input in the model and the aim is to locate a given number of facilities. One exception is the set covering location model, where the aim is to minimize the number of service facilities needed to cover orders from all demand facilities within a given distance. In this case, the number of service facilities is determined endogenously. Another type of model where the number of service facilities is determined endogenously is the model of fixed charge facility location problems, where the objective function contains information about fixed costs and variable costs to open and operate a service facility.

We will now look at solution technique approaches. Classical approaches of bivalent programming and enumeration methods can be used to solve location models. However, in recent

literature, various heuristics and metaheuristic algorithms are often used which, in a relatively short time, find close to optimal solutions. Such heuristics include, for example, Lagrange heuristics [53], greedy heuristics, tabu search, simulated annealing [54], genetic algorithms [55], evolutionary algorithm, bee colony algorithm [56], and ant colony optimization. In this paper, an evolutionary algorithm was used due to the size of the task and the fact that it is not possible to use classic approaches. Differential evolution [57] is considered to be an appropriate tool for the solution. Differential evolution (DE) is a stochastic algorithm for solving numerical optimization problems. Since its inception, the DE algorithm has become a powerful global optimizer.

This paper is organized as follows. Section 2 describes the mathematical programming model formulation. As the problem is intractable in terms of classical optimization techniques, a modified formulation of the problem design for the metaheuristics approach is described. Section 3 introduces the metaheuristic approach—a differential evolution computation algorithm for the location of *Osmia* nesting aids. Sections 4 presents results, specifically the optimal site of nesting aids in an orchard with randomly distributed trees, or orchards with various shapes, i.e., square, L-shaped, or X-shaped to demonstrate an optimal solution for different case studies where the pollination of fruit trees is needed. Finally, Section 5 presents our conclusions.

## 2. Mathematical programming models for determining the optimal location of nesting aids

Suppose that fruit trees requiring pollination are randomly located, and the distances between the trees are known. Assume that the nesting aids, i.e. hives can be located under any tree, and the total number of nesting aids is given. The aim is to pollinate all the trees at a minimum total distance flown by the solitary bees. The fundamental location problem is known as the *p*-median problem [37] and the set covering problem [42,43].

To solve the introduced mathematical programming problem of locating the nesting aids in the orchard, we have set two objectives: the first objective is to minimize the number of nesting aids. The second objective aim is to minimize the total distance flown by all solitary bees. In this case, we will discuss lexicographic multi-objective linear programming, as the priorities of each objective function are different. When dealing with multiple criteria programming problems, we assume two types of relationships between objective functions: the relationship between the importance of the criteria is defined by their ratio or, as in our case, by setting their priorities. The theory and algorithm of lexicographic multi-objective linear programming problems can be found in [58].

The problem can be formulated as a binary programming problem with variables $y_{ij} \in \{0, 1\}$, $x_i \in \{0, 1\}$, $i, j = 1, 2, \ldots n$, where $n$ represents the number of trees in the orchard. The variables $y_{ij} \in \{0, 1\}$, $i, j = 1, 2, \ldots n$ represent the pollination event. The value is equal to 1 if the *j*-th tree is pollinated by solitary bees whose origin is from the *i*-th nesting aid, otherwise, the value is equal to 0. The model also deals with binary variables $x_i \in \{0, 1\}$, $i = 1, 2, \ldots n$ that represent the location of the nesting aids. The value is equal to 1 if a nesting aid is operated under the *i*-th tree, otherwise, it is equal to 0.

Suppose there are goals with different priorities. The highest priority is assigned to the goal of minimizing the number of nesting aids. Let us assume a second priority level goal, i.e., minimizing the total distance flown by all solitary bees. The objective deals with parameters $d_{ij}$, $i$, $j = 1, 2, \ldots n$ that represent minimum distances between all trees in the orchard. The aim is to pollinate all the trees. Notations P1 and P2 denote the priority level, and objective functions of

the location problem can be formulated:

$$\text{P1 } f_1(\mathbf{x}, \mathbf{y}) = \sum_{i=1}^{n} x_i \rightarrow \min \tag{1}$$

$$\text{P2 } f_2(\mathbf{x}, \mathbf{y}) = \sum_{i=1}^{n} \sum_{j=1}^{n} d_{ij} y_{ij} \rightarrow \min \tag{2}$$

The first constraint represents the need to pollinate all the trees from the $i$-th location of the nesting aid.

$$\sum_{i=1}^{n} y_{ij} = 1, j = 1, 2, \dots n \tag{3}$$

The second constraint models the maximum distance the solitary bee can fly. Setting the parameter $K$ to a "worst-case" value will ensure pollination in the event of adverse weather conditions.

$$\sum_{i=1}^{n} d_{ij} y_{ij} \leq K, j = 1, 2, \dots n \tag{4}$$

The constraints that represent the need to locate the nesting aid at the $i$-th location belong to the third constraint.

$$y_{ij} - x_i \leq 0, i, j = 1, 2, \dots n \tag{5}$$

It is also evident that the number of nesting aids must be restricted by the first priority objective function that ensures the total number of nesting aids $p^*$ is minimized:

$$\sum_{i=1}^{n} x_i = p^* \tag{6}$$

The formulation of the mathematical programming model on the first priority level is a modified set covering problem:

$$
\begin{aligned}
f_1(\mathbf{x}, \mathbf{y}) \quad &= \sum_{i=1}^{n} x_i \rightarrow \min \\
&\sum_{i=1}^{n} y_{ij} = 1, j = 1, 2, \dots n \\
&\sum_{i=1}^{n} d_{ij} y_{ij} \leq K, j = 1, 2, \dots n \\
&y_{ij} - x_i \leq 0, i, j = 1, 2, \dots n \\
&x_i, y_{ij} \in \{0, 1\}, i, j = 1, 2, \dots n
\end{aligned}
\tag{7}
$$

Let the optimal value of the total number of nesting aids be $p^* = \sum_{i=1}^{n} x_i^*$, we can then formulate the mathematical programming model at the second priority level as a modified $p$-median

problem as follows:

$$f_2(\mathbf{x}, \mathbf{y}) = \sum_{i=1}^{n}\sum_{j=1}^{n} d_{ij} y_{ij} \rightarrow \min$$

$$\sum_{i=1}^{n} y_{ij} = 1, j = 1, 2, \ldots n$$

$$\sum_{i=1}^{n} d_{ij} y_{ij} \leq K, j = 1, 2, \ldots n \tag{8}$$

$$y_{ij} - x_i \leq 0, i, j = 1, 2, \ldots n$$

$$\sum_{i=1}^{n} x_i = p^*$$

$$x_i, y_{ij} \in \{0, 1\}, i, j = 1, 2, \ldots n$$

Where:

$n$ represents the number of trees in the orchard,

$y_{ij}$–represents the pollination event of the $j$-th tree by bees from the $i$-th nesting aid,

$x_i$-represents the location of the nesting aid under the $i$-th tree. It is assumed that the location of a nesting aid can be under any tree,

$d_{ij}$-represents the shortest distance from the $i$-th nesting aid to the $j$-th tree, or from the $i$-th tree to the $j$-th tree.

$p^*$–represents the number of nesting aids that need to be installed,

$K$–represents the maximum distance the solitary bee can fly.

During the solution implementation step, we encounter problem solvability, as the number of variables is $n^2 + n$. A real-life problem, detailed in the fourth part of the article, concerns the location of nesting aids in an orchard of one hectare (10,000 m$^2$), for standard fruit tree distances, i.e., for 2,184 trees and thus 4,772,040 variables. The above problem is intractable using classical optimization techniques because of the number of variables. It was, therefore, necessary to reduce the number of variables by formulating an equivalent model. The reduction in the number of variables could be achieved by introducing constraints (10–12) that are not suitable for classical optimization tools. It reduced the number of variables to twice the number of nesting aids -2 $p$. For this reason, we modify the formulation of the problem as follows:

Let the coordinates of the individual trees be known and be represented by the [$a_j$, $b_j$], where $a_j$ represents the first coordinate of the $j$-th tree and $bj$ the second coordinate of the $j$-th tree. Let the number of trees be $n$. Let $M$ be a large positive number. Parameter $d$ represents the maximum bee foraging range. The model $x_i$ variables represent the serial number of the tree on which the nesting aid is located and the maximum number of nesting aids in the orchard is represented by $m$. In addition to the above variables, the model contains the binary variables $z_k \in \{0, 1\}$, $k = 1, 2, \ldots m$, which can have two values. Value 1 indicates the nesting aid is placed on the tree, and value 0 indicates the nesting aid is not placed on the tree.

As in the previous model formulation, we must define objective functions for each priority level. At the first priority level, the objective function P1 represents the total number of nesting aids used, the value of which is represented by the sum of the variables $z_k \in \{0, 1\}$, $k = 1, 2, \ldots m$. The second priority objective function models the total distance flown by all the bees to the individual trees. Based on the above statement, the objective function P2 is the sum of

Euclidian distances between the trees to the nearest nesting aid.

$$P1 \quad \min f_1(\mathbf{x}, \mathbf{z}) = \sum_{i=1}^{m} z_i$$

$$P2 \quad \min f_2(\mathbf{x}, \mathbf{z}) = \sum_{j=1}^{n} \min_{i=1,2,\ldots m} \left\{ \sqrt{(a_j - a_{x_i})^2 + (b_j - b_{x_i})^2} \right\}$$

(9)

The group of first structural constraints ensures that the nearest tree on which the nesting aid is located must be at a maximum distance $d$.

$$\min_{i=1,2,\ldots m} \left\{ (1 - z_i) \cdot M + \sqrt{(a_j - a_{x_i})^2 + (b_j - b_{x_i})^2} \right\} \leq d, j = 1, 2, \ldots n \qquad (10)$$

Since we assume the location of the nesting aid is on a tree, we must ensure that its distance from the nearest tree is equal to 0 for the used nesting aid in the orchard.

$$z_i \cdot \min_{j=1,2,\ldots n} \left\{ \sqrt{(a_j - a_{x_i})^2 + (b_j - b_{x_i})^2} \right\} = 0, i = 1, 2, \ldots m \qquad (11)$$

The last group of constraints ensures the variables $z_k$ ($k = 1,2,\ldots m$) take on value 1 for nesting aids used in the orchard.

$$M \cdot \left( \sqrt{(a_j - a_{x_k})^2 + (b_j - b_{x_k})^2} - \min_{i=1,2,\ldots m} \left\{ \sqrt{(a_j - a_{x_i})^2 + (b_j - b_{x_i})^2} \right\} \right) + z_k \geq 1, j$$

$$= 1, 2, \ldots n, k = 1, 2, \ldots m \qquad (12)$$

Based on the above, the modified problem can be formulated as follows:

$$P1 \quad \min f_1(\mathbf{x}, \mathbf{z}) = \sum_{i=1}^{m} z_i$$

$$P2 \quad \min f_2(\mathbf{x}, \mathbf{z}) = \sum_{j=1}^{n} \min_{i=1,2,\ldots m} \left\{ \sqrt{(a_j - a_{x_i})^2 + (b_j - b_{x_i})^2} \right\}$$

$$\min_{i=1,2,\ldots m} \left\{ (1 - z_i) \cdot M + \sqrt{(a_j - a_{x_i})^2 + (b_j - b_{x_i})^2} \right\} \leq d, j = 1, 2, \ldots n$$

$$z_i \min_{j=1,2,\ldots n} \left\{ \sqrt{(a_j - a_{x_i})^2 + (b_j - b_{x_i})^2} \right\} = 0, i = 1, 2, \ldots m \qquad (13)$$

$$M \cdot \left( \sqrt{(a_j - a_{x_k})^2 + (b_j - b_{x_k})^2} - \min_{i=1,2,\ldots m} \left\{ \sqrt{(a_j - a_{x_i})^2 + (b_j - b_{x_i})^2} \right\} \right)$$

$$+ z_k \geq 1, j = 1, 2, \ldots n, k = 1, 2, \ldots m$$

$$x_i \in \{0, 1, \ldots n\}, i = 1, 2, \ldots m, z_k \in \{0, 1\}, k = 1, 2, \ldots m$$

Let the optimal value of the first priority level be equal to $p^* = \sum_{i=1}^{m} z_i^*$ then, on the second priority level, we set the value of the maximum number of nesting aids to $m = p^*$.

Classical optimization techniques cannot solve the model (13) because of the atypical structure of constraints. Based on the above, the metaheuristics approach was used to find a solution to the problem. Differential evolution has been applied based on the authors' previous work [59] from the many well-known metaheuristic approaches [60,61]. Problems (7), (8), and (13) were solved based on optimization techniques and differential evolution for small

numbers of variables. The obtained results were identical, which is based on the nature of the formulation of models.

## 3. Differential evolution for the location of *Osmia* nesting aids

Differential evolution is an evolutionary computation algorithm. It is a metaheuristics method that optimizes a problem by iteratively attempting to improve a candidate solution within vast spaces of candidate solutions. The computational complexity of the presented location problem arises from its nonlinear structure. Therefore, evolutionary algorithms are considered to be a suitable alternative to standard techniques, due to their ability to achieve suboptimal solutions in a relatively short time. However, metaheuristics such as differential evolution do not guarantee an optimal solution will be found.

Differential evolution, introduced by Storn and Price [62], is an evolutionary technique, which forms the basis of many nontraditional computing techniques whose common characteristic is they are based on the observation of natural processes. Evolutionary algorithms are practical tools that can be used to search for solutions to a wide range of optimization problems, e.g., [59,63–67]. The advantage over traditional optimization methods is that they are designed to find global extremes. Their use does not require a priori knowledge of optimized functions, such as convexity, or differential functions. With few or no assumptions about the optimized problem, they are effective in solving continuous nonlinear problems, where it is difficult to use traditional mathematic methods.

Differential evolution involves a search of a population of individuals (*np*–number of individuals). Each individual represents a candidate solution for the given problem represented by parameters of the individual (*d*–number of parameters). The fitness ($f_{cost}$) is associated with each individual, representing the objective function's relevant value. Every step of the algorithm involves a competitive selection that carried out weak solutions. The algorithms' steps are described in detail, e.g. [68,69], and the principle of the basic version of differential evolution can be characterized by the pseudocode listed in [63,65].

The following factors need to be considered before applying a differential evolution algorithm to solve the above-defined location of *Osmia* nesting aid problems: *selecting an appropriate representation of individuals*, *setting control parameters*, *and transformation of infeasible solutions* [65]. It is possible to summarize the principle of the algorithm as follows:

*Select an appropriate representation of individuals*. For the location of *Osmia* nesting aids, it is appropriate to use the following description of an individual in each generation (let *s* be the index corresponding with the number of the generation, so that *s* = 0, 1,. . .*g*). Each generation is represented by the matrix $\mathbf{X}^{(s)}$, which consists of *np* individuals represented by the vectors $\mathbf{x}_i^{(s)}$ (order number of the tree on which a nesting aid is located), *i* = 1, 2, . . .*np*, where each parameter of the individual represents the corresponding position of the location aid $x_{i,j}^{(s)}, i = 1, 2, \ldots np, j = 1, 2, \ldots d$. The use of the stated representation involves a simple calculation of fitness ($f_{cost}(\mathbf{x}_i^{(0)})$, *i* = 1, 2, . . .*np*) using the optimization function at the first and second priority levels (8) defined in the second part of the article.

*Set control parameters*. A particular set of parameters controls the differential evolution.

*d–dimensionality*. The number of parameters of an individual is equal to the maximum number of location aids.

*np–population size*. The number of individuals in a population. The recommended setting is from *5d* to *30d*, or *100d*, where the optimized function is multimodal [70,71].

*g–generations*. The maximum number of iterations (*g* is also the stopping criterion) can be set based on experiments.

*cr–crossover constant*. The value of *cr* can be set based on experiments. The crossover constant *cr* should not be too large to avoid the perturbations getting too high and the convergence speed decreasing. However, a small *cr* decreases diversity and might cause the strategy to get stuck [72].

*f–mutation constant*. The value of *f* can be set based on experiments. The mutation constant should not be smaller than a specific value to avoid the population converging before arriving at the minimum. On the other hand, the mutation constant should not be chosen too large because the number of function evaluations increases as the value increases [72].

Recommended values for the parameters are usually derived empirically from experiments described in, e.g., [69,71,73], or statistical methods can be applied, which are described in, e.g., [74,75].

*Initialization*. The initial population $\mathbf{X}^{(0)}$ can be randomly initialized at the beginning of the evolutionary process according to the rule:

$$x_{ij}^{begin} = rnd\langle 0, \max\rangle_i = 1, 2, \ldots np \; j = 1, 2, \ldots d$$
$$x_{ij}^{(0)} = round(x_{ij}^{begin}, 0) \; i = 1, 2, \ldots np \; j = 1, 2, \ldots d$$

where $rnd\langle 0, \max\rangle$ represents random numbers that are drawn from a uniform distribution in an interval from zero to the defined maximum. The round function is used due to the need to place a nesting aid on one of the trees. The above rules ensure that each tree must be within 50 meters of the nearest nesting aid. Each individual is then evaluated with $f_{cost}(\mathbf{x}_i^{(0)})$, $i = 1, 2, \ldots np$.

*The test of stopping condition*. The only stopping criterion is that the maximum number of iterations (represent by parameter *g*) is reached.

*Set the transformation of infeasible solutions*. The reproductive cycle comprises crossing and mutation to create individuals for the next generation. Infeasible solutions are subject to the following rule, which ensures the feasibility of the reproductive cycle solution:

If $x_{ij}^{test} \neq round(x_{ij}^{test}, 0)$, then $x_{ij}^{Finaltest} = round(x_{ij}^{test}, 0)$

The objective function's test vector value is compared to the vector value of the objective function of the currently selected individual, and the next generation is selected based on the vector, with a better objective function value, formally:

$$\mathbf{x}_i^{(s+1)} = \left\{ \begin{array}{l} \mathbf{x}_i^{FinalTest}, \text{if } f_{cost}(\mathbf{x}_i^{FinalTest}) \leq f_{cost}(\mathbf{x}_i^{(s)}) \\ \mathbf{x}_i^{(s)}, \text{otherwise} \end{array} \right\}$$

So this process continues in each generation for all individuals and the result is a new generation $\mathbf{X}^{(s+1)}$ with the same number of individuals.

*Evaluation of calculating process*. The reproduction process continues until the last (users specified) number of generations is reached. The value of the best individual from each generation is reflected in the *history vector* and this shows the progression of the evolutionary process.

## 4. Location of nesting aids in an orchard with randomly placed trees

Random tree locations were used to test the solution of the nesting aids location problem, with a precondition for determining the minimum spacing between the trees. The minimum distance between the trees in a row was 1.2 m and in a column 4 m. Another assumption was the location of nesting aids in a row on a tree, so in the orchard nesting aids may only be located

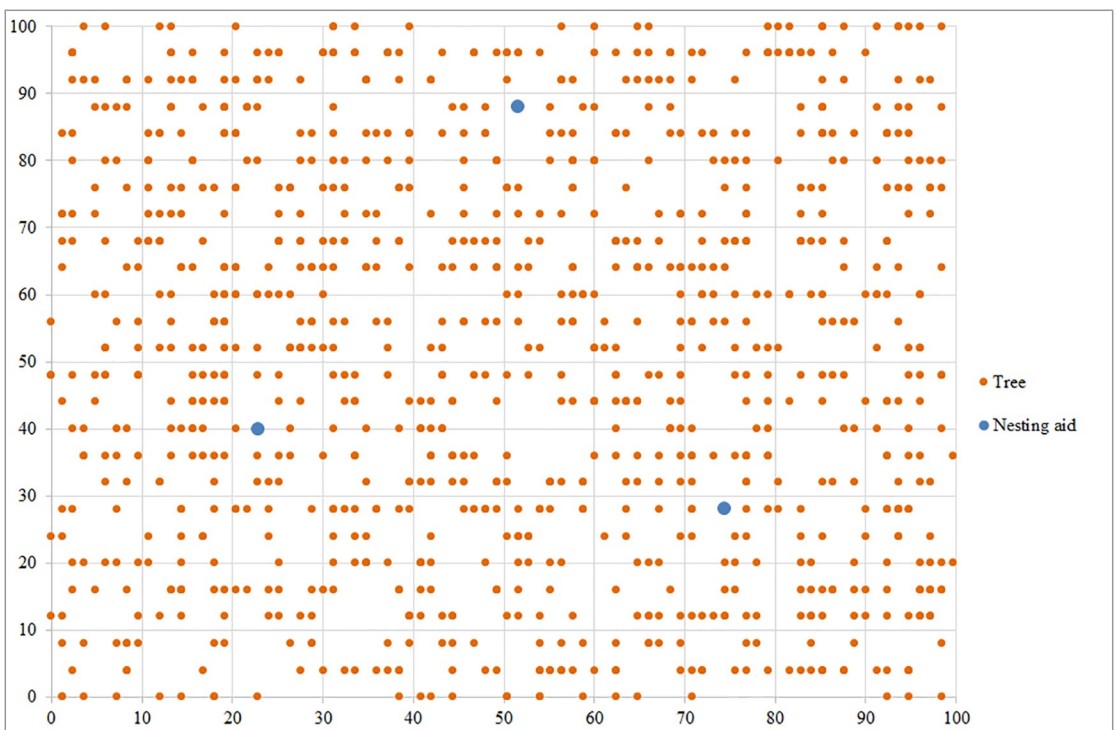

**Fig 1. The optimal solution for an illustrative example with 1,000 randomly located trees that need to be pollinated.** Source: authors' own calculations.

on a tree, or under a tree. In addition to the above, the maximum foraging range of *Osmia* bees is set to 50 m. The area of the orchard was set to 10,000 m$^2$ (100x100 m).

We show the results for a randomly generated set of trees in an orchard in Fig 1 to demonstrate the universality and applicability of the proposed approach for any collection of trees. The orange dots represent trees. Using the model set out in the second part of this paper, which we solved with the algorithm described in part three, we obtained the presented results. After solving the task at the first priority level, we obtained the minimum number of nesting aids needed to achieve pollination of all the trees, and in our case, it was found to be necessary to install three nesting aids. When addressing the task at the second priority level, we were given the final location of nesting aids depicted in Fig 1 by the blue dots, where the aim was to minimize the total distance flown by *Osmia* bees.

For the desired objectives, it is necessary to place individual nesting aids on trees in the following positions: the first nesting aid is located on the tree with coordinates (51.6 m, 88 m), the second nesting aid on the tree with coordinates (22.8 m, 40 m) and the third nesting aid on the tree with coordinates (74.4 m, 28 m). The total distance flown by *Osmia* bees is 24,109.57 m.

The solution of the nesting aids location problem was implemented in R using the DEoptim library [76]. R is a programming language and environment designed for data analysis and graphical display. R is a freely available language and environment for statistical computing and graphics, which provides a wide variety of statistical and graphical techniques: linear and nonlinear modeling, statistical tests, time-series analysis, classification, clustering. Based on the general procedure of the differential evolution algorithm set out in the previous part of the article, we implemented the solution of the problem stated above in software R.

In the solution procedure, it was necessary to set the values of the differential evolution algorithm's parameters, and the values were set based on experiments conducted in previous research, according to [63,74]:

*d–dimensionality*. The number of individual parameters is equal to the maximum possible number of nesting aids, which was set to 10.

*np–population size*. The number of individuals in the population was set to 100.

*g–generations*. The maximum number of iterations was set to 1,000.

*cr–crossover constant*. The value of *cr* was set to 0.2.

*f–mutation constant*. The value of *f* was set differently for two priority levels: for the first priority level it was set to 0.9. A higher mutation rate is needed to obtain the solution. The second priority level was set to 0.1. A lower mutation rate is required, as a solution to the minimum number of nesting aids was obtained in the first phase. Subsequently, a second priority level solution must be found from the area around the solution identified in the first phase. These values were obtained based on the tests carried out for the different mutation constant values, and the main criterion was the speed of convergence.

In addition to the above values, we set the lower and upper bound of variables in DEoptim v R.

*lb*—lower bound–The values of each variable were set to 0.

*ub*—upper bound–The values of each variable were set to 1000 (number of trees).

When addressing the task at the first priority level, we obtained a value for the minimum number of nesting aids, which was a value of three. After setting the value of the minimum number of nesting aids that need to be located, we subsequently addressed the task at the second priority level. Fig 2 shows the convergence plot of the objective function value at the second priority level, which gradually converged to 24,109.57.

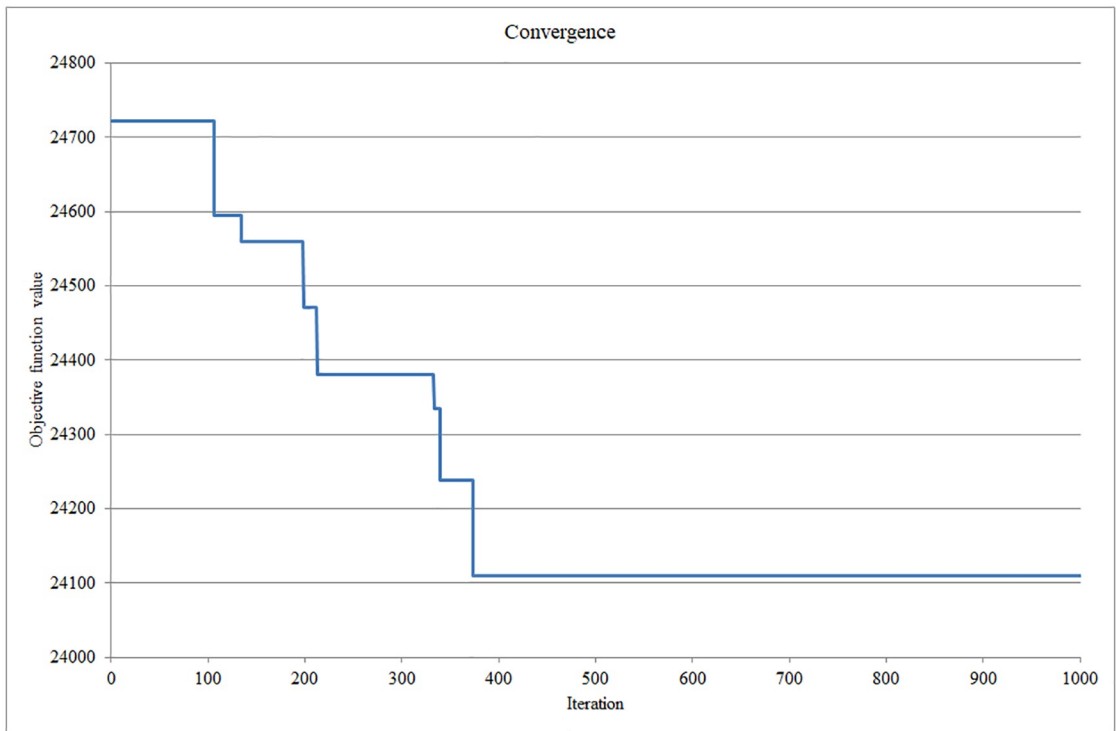

**Fig 2. Convergence of the objective function value at the second priority level model.** Source: authors' own calculations.

## 5. Case study

In the next section, we illustrate the application of the nesting aids location problem for various orchard shapes, i.e., square, L-shaped, or X-shaped to demonstrate an optimal solution for different case studies where the pollination of fruit trees is required. Typical orchards tend to be of different shapes, usually with fixed distances between trees, unlike the example in chapter four, with randomly spaced trees. Therefore, we will further analyze the different shapes of orchards with regularly spaced trees. All the analyzed shapes of orchards have an area of 10,000 m$^2$. The simulated orchards follow the spatial pattern used by farmers: the distance between the trees in a row was set to 1.2 m and in a column to 4 m, according to [16]. The solitary bee *O. cornuta* is unselective regarding its nesting habitat, and frequently colonizes artificial nests [77]. Assumptions, such as the location of an artificial nesting aid on a tree, or under a tree, and the maximum foraging range of *Osmia* bees within 50 m, were used. Applying a maximum foraging range of 50 m enables modeling of pollination to ensure it occurs in the event of adverse weather conditions. Although the foraging ranges of *O. cornuta* females are higher than the parameter set in our study, [15,26] states a forage distance within 200 m of the nesting site. A more recent study [78] published data on flight distances of tagged solitary bees in the Munich Botanical Garden in 2017 and 2018. This gave female *O. cornuta* a mean flight distance of 107 m, with a standard deviation 67.9 m, and a maximum flight distance of 226 in 2018. A maximum foraging range of 50 m enables us to model for the fact that a higher density of nesting aids represents a greater density of foraging and thus higher pollination activity, as pollen is collected to fill each nest in a nesting aid [79]. In the paper [80], the authors model the minimization of the bees' flight stress by making the maximum flight distance assumption stricter In order to perform calculations in R, it is necessary to specify not only the input parameters but also the parameters of differential evolution, which are in all three cases identical to the setting of the parameters from chapter four, except the parameter upper bound, thus $d = 10$; $np = 100$; $g = 1,000$; $cr = 0.2$; $f = 0.9$ for the first priority level; $f = 0.1$ for the second priority level and $lb = 0$; and $ub_{square} = 2,184$; $ub_{L\text{-}shaped} = 2,154$; $ub_{X\text{-}shaped} = 2,146$.

### A) Location of nesting aids in a standard planted set of fruit trees in a square-shaped orchard

We illustrate the results in a standard planted set of fruit trees in a square-shaped orchard in Fig 3. The orange dots represent trees. After solving the task at the first priority level, we obtained the minimum number of nesting aids needed to achieve pollination of all the trees. In this case, as opposed to the randomly generated location of trees, four nesting aids must be installed. After solving the task at the second priority level, we obtained the nesting aids' final position depicted in Fig 3 by blue dots. The goal was to minimize the total flying distance of the bees.

For the desired objectives, it is necessary to place individual nesting aids on trees in the following positions: the first nesting aid is located on the tree at coordinates (24, 76), the second nesting aid on the tree at coordinates (74.4, 76), the third nesting aid on the tree at coordinates (24, 24) and the fourth nesting aid on the tree at coordinates (75.6, 24). The total distance flown by *Osmia* bees is 42,740.96 m.

### B) Location of nesting aids in orchards with various shapes

In the previous section, we envisaged a standard planted set of fruit trees in a square-shaped orchard. This section will detail the application of the above approach to different types of orchard shapes, with an area of 10,000 m$^2$. The spacing between the trees and foraging distance

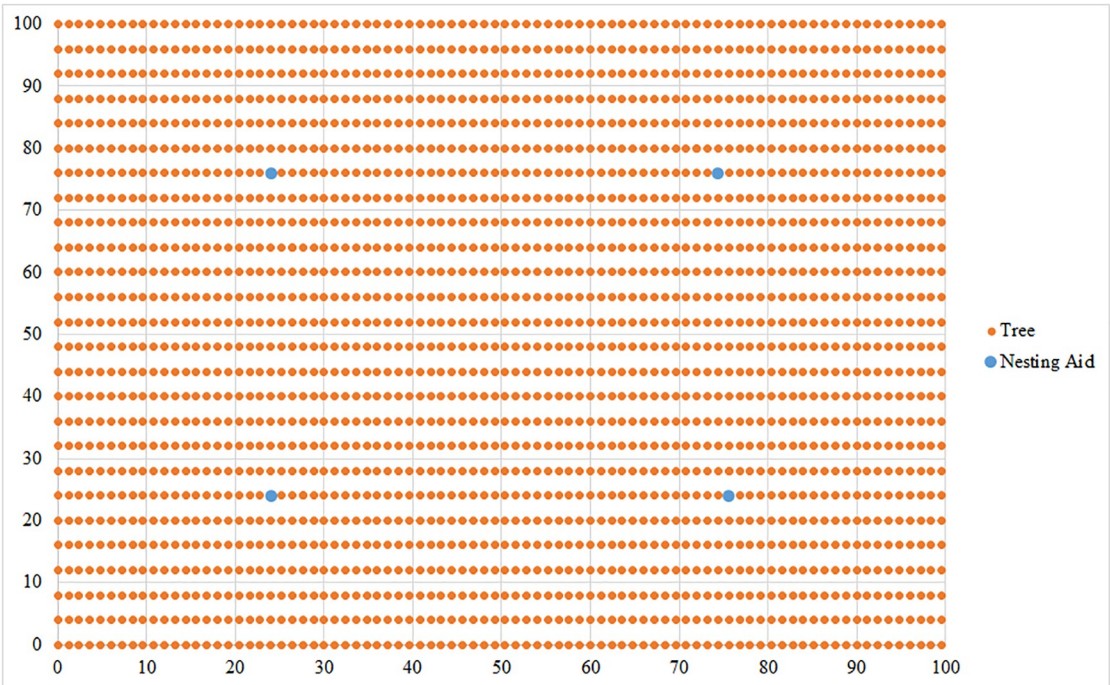

**Fig 3. The optimal solution for a standard planted set of fruit trees in a square-shaped orchard with 2,184 trees that require pollination.** Source: authors' own calculations.

assumption remains the same as in the previous example. We will illustrate the application of the nesting aids location problem for L-shaped and X-shaped orchards to demonstrate an optimal solution for the location of nesting aids where the pollination of fruit trees is required.

The setting and final results for the L-shaped orchard are shown in Fig 4. After solving the task at the first priority level, we obtained the minimum number of nesting aids needed to pollinate all the trees. In this case, it is necessary to install five nesting aids. When addressing the task at the second priority level, we were given the nesting aids' final position depicted in Fig 4 by blue dots. The goal was to minimize the total flying distance of the bees.

For the desired objectives, it is necessary to place individual nesting aids on trees in the following positions: the first nesting aid is located on the tree with coordinates (26.4, 60), the second nesting aid on the tree with coordinates (37.2, 16), the third nesting aid on the tree with coordinates (92.4, 72), the fourth nesting aid on the tree with coordinates (172.8, 72) and the fifth nesting aid on the tree with coordinates (252, 72). The total distance flown by *Osmia* bees is 45,320.8 m. The results are shown in Fig 4. The blue dots represent the optimal location of nesting aids in the orchard and trees that need to be pollinated are depicted by orange dots.

The setting and final results for the X-shaped orchard are shown in Fig 5. After solving the task at the first priority level, we obtained the minimum number of nesting aids needed to pollinate all the trees. In this case, it is necessary to install four nesting aids. When addressing the task at the second priority level, we were given the nesting aids' final position depicted in Fig 5 by blue dots. The goal was to minimize the total flying distance of the bees.

To achieve the desired objectives, it is necessary to place individual nesting aids on trees in the following positions: the first nesting aid is located on the tree with coordinates (22.8, 108), the second nesting aid on the tree with coordinates (27.6, 36), the third nesting aid on the tree with coordinates (60, 160) and the fourth nesting aid on the tree with coordinates (70.8, 112).

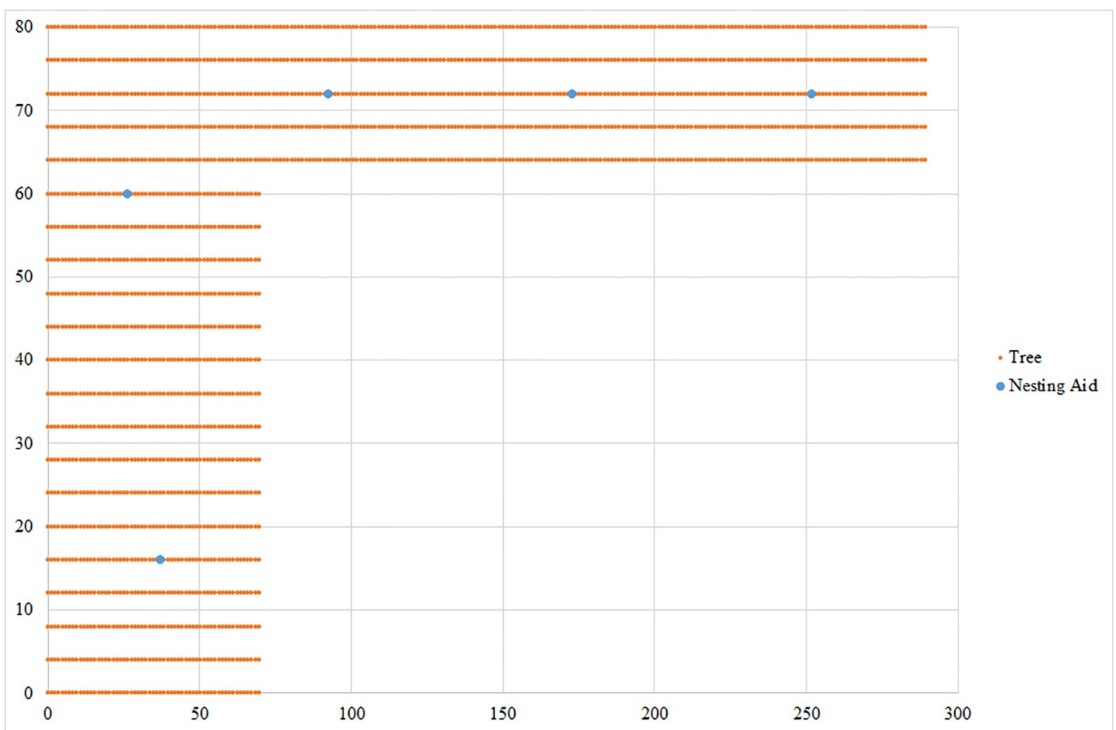

**Fig 4. The optimal solution for an L-shaped orchard with 2,154 trees that need to be pollinated.** Source: authors' own calculations.

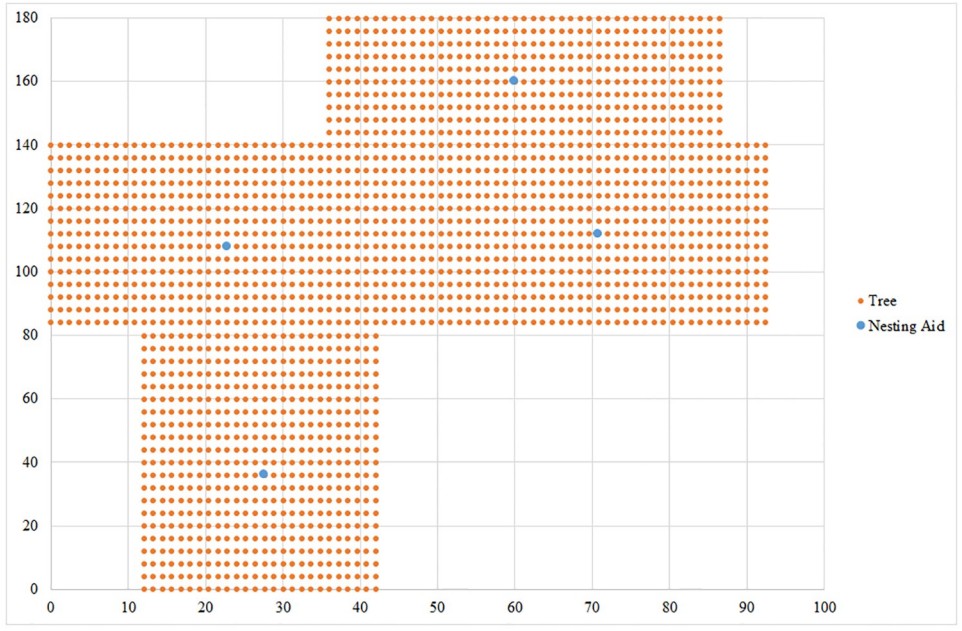

**Fig 5. The optimal solution for a standard X-shaped orchard with 2,146 trees that need to be pollinated.** Source: authors' own calculations.

The total distance flown by *Osmia* bees is 43,619.6 m. The results are shown in Fig 5. The blue dots represent the optimal location of nesting aids in the orchard and trees that need to be pollinated are represented by orange dots.

## Conclusion

Pollination is an economically important ecosystem service that is threatened by biodiversity losses [10]. Agricultural intensification undermines pollinator abundance and diversity. Insufficient pollination is a common cause of low yields. Most agriculturally dominated landscapes are impoverished and represent a hostile environment for solitary bees, despite mass flowering crops. The management of bee species is key to crop production, especially where wild bees are present in low numbers, such as intensive crop production fields with less uncultivated lands in the surroundings [81]. Honey bees can be used as pollinators in large commercial orchards to improve productivity [13], but several other insects, notably solitary bees have been studied as alternative pollinators and have been demonstrated to be effective pollinators of, for example, apple, cherry, pear, and strawberry crops and in some cases have been found to be more effective than honeybees [14]. Advantages of alternative pollinators are the low temperatures when such pollinators remain active, their preference for foraging on flowers of the target crop and, stigma contact by floral visitors, resulting in higher yields and quality and thus economic output. Several alternative bee species have been found to be able to replace or supplement honey bees. For example, Bosch and Kemp [31] looks at establishing and managing *Osmia* solitary bees species as crop pollinators of fruit trees.

The contribution of insect pollinators to agricultural output is evident, but obviously there are costs involved with the management of pollinators. This paper seeks to contribute to the management of pollinators by providing a quantitative decision tool to support decisions regarding the optimal location of nesting aids. The paper presents a mathematical programming model that seeks to find the optimal location for the smallest number of nesting aids for solitary bees in an orchard, while ensuring all trees are pollinated and a minimum total distance is flown by the insects.

The proposed model's application is illustrated on the nesting aids location problem for various orchard shapes, e.g., square, L-shaped, or X-shaped to demonstrate an optimal solution for different case studies where the pollination of fruit trees is required. The location results are shown in the Figs 3–5. The blue dots represent the optimal location of nesting aids in an orchard and trees that need to be pollinated are represented by orange dots. All the analyzed shapes of orchards have the same area of 10,000 m$^2$. Simulated orchards follow the spatial pattern used by farmers: the distance between the trees in the row was set to 1.2 m, and in the column to 4 m, according to [16]. Assumptions such as the location of an artificial nesting aid on the tree, or under a tree, and the maximum foraging range of *Osmia* bees were used. Applying a maximum foraging range enables the modeling of pollination to ensure it still occurs in the event of adverse weather conditions, and enables the modelling of the fact that a higher density of nesting aids represents a greater density of foraging and thus higher pollination activity, or minimizes the bees' flight stress by making the maximum flight distance assumption stricter.

The proposed location model is original and unique. Instead of a random ad hoc location of nesting aids in an orchard, or at the edge of an orchard, it utilizes an effective optimization tool to determine locations which will optimize pollination by alternative pollinators, and increase the economic output of an agricultural business. The importance of this proposed model is likely to increase with agricultural intensification, and the decrease of the numbers of wild pollinators.

## Supporting information

**S1 Data. Coordinates of the trees.**
(XLSX)

## Author Contributions

**Conceptualization:** Juraj Pekár, Marian Reiff, Ivan Brezina.

**Data curation:** Juraj Pekár, Marian Reiff, Ivan Brezina.

**Formal analysis:** Juraj Pekár, Marian Reiff, Ivan Brezina.

**Funding acquisition:** Juraj Pekár, Marian Reiff, Ivan Brezina.

**Investigation:** Juraj Pekár, Marian Reiff, Ivan Brezina.

**Methodology:** Juraj Pekár, Marian Reiff, Ivan Brezina.

**Project administration:** Juraj Pekár, Marian Reiff, Ivan Brezina.

**Resources:** Juraj Pekár, Marian Reiff, Ivan Brezina.

**Software:** Juraj Pekár, Marian Reiff, Ivan Brezina.

**Supervision:** Juraj Pekár, Marian Reiff, Ivan Brezina.

**Validation:** Juraj Pekár, Marian Reiff, Ivan Brezina.

**Visualization:** Juraj Pekár, Marian Reiff, Ivan Brezina.

**Writing – original draft:** Juraj Pekár, Marian Reiff, Ivan Brezina.

**Writing – review & editing:** Juraj Pekár, Marian Reiff, Ivan Brezina.

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
