## [Decision Letter · Decision Letter 0]

11 Nov 2020

PONE-D-20-30891

Location problem of Osmia cornuta nesting aids for optimum pollination

PLOS ONE

Dear Authors,

Thank you for submitting your manuscript to PLOS ONE. After careful consideration, we feel that it has merit but does not fully meet PLOS ONE’s publication criteria as it currently stands. Therefore, we invite you to submit a revised version of the manuscript that addresses the points raised during the review process.

 Please see comments below.

We look forward to receiving your revised manuscript.

Kind regards,

Dejan Dragan, PhD

Academic Editor

PLOS ONE

Additional Editor Comments:

AE's decison: Major revision. The article was reviewed by two reviewers. One of them requires a major revision, while the other requires the minor revision. Therefore, it is suggested that the authors strictly follow the instructions and comments of the reviewers.

Journal Requirements:

Reviewers' comments:

Reviewer's Responses to Questions

**Comments to the Author**

1. Is the manuscript technically sound, and do the data support the conclusions?

Reviewer #1: Yes

Reviewer #2: Partly

2. Has the statistical analysis been performed appropriately and rigorously? 

Reviewer #1: No

Reviewer #2: N/A

3. Have the authors made all data underlying the findings in their manuscript fully available?

Reviewer #1: Yes

Reviewer #2: Yes

4. Is the manuscript presented in an intelligible fashion and written in standard English?

Reviewer #1: Yes

Reviewer #2: Yes

5. Review Comments to the Author

Reviewer #1: The proposed methodology for achieving the stated objective is well described facing formal aspects and illustrating by means of a real data application. The background and references include all the works treating the issues faced in this paper. The work is well-structured starting from a description of some preliminary concepts, a review of the state of the art and then proposing an improved methodology.

Regarding introduction:

Once the introduction is read, the reader can guess the main objective, but it seems to be several the specific objectives to get the main one. The problem is that is not written clearly nor highlighted in the introduction what is this main goal. In fact, a reader like me identify the main goal at line 90, but the introduction follows another 76 lines with an important and detailed review (I like this) of the models and different solution approaches. Perhaps would be more appropriated this review if it were more concise and at the beginning of the next section, even dividing at the beginning of the two next sections.

Regarding section two:

The authors suppose randomly located trees but distances between then are known (in a compact set I guess). I do not know if the consideration of random distances would be of interest, since the first objective (line 175) is to minimize the nesting aids and. In my opinion, random locations but pre-fixed distances is not so random, surely uniformly randomized distributed. I apologize in advance if I am confused.

On the other hand the formulations of the problem in equations (1), (2), (7) and (8) would be more illustrative if this equations write the minimum in what (as they do in equations (10),(11) and (13)), not only min, because this notation forces the reader going reading back.

In line 232 the authors affirm that the problem is intractable with classical optimization techniques. A short sentence explaining why would be of interest.

Finally, once the original problem is rewritten by means of a metaheuristic approach, may be of interest to justify how this formulation is equivalent to the first one, i.e. the solution of the second provides the solution of the first one.

Regarding section three:

To my knowledge, there are different metaheuristics approaches, each one with its advantages and disadvantages. Perhaps a short sentence that justify this choice would be enrichment.

I enjoyed the reading of this manuscript because everything is very detailed, and in my opinion, this approach has a direct applicability. If the authors consider some of my recommendations these may be improve the reading of manuscript.

Anyway, my decision would be to accept for publication.

Reviewer #2: The authors of “Location problem of Osmia cornuta nesting aids for optimum pollination” propose a practical solution for optimisation of bees nesting aids locations. The idea of usage of differential evolution algorithm for this purpose is interesting and promising, very well explained and therefore motivated in the first three chapters of the proposed text. I think that minor revision/clarifications are required there. However, the experimental part is not very supportive and some of the claims are not clearly demonstrated in the following chapters. All my recommendations and questions are listed below:

1. At first, the citation of R and ‘Deoptim’ library is not complete. The R version must be cited as it is shown in the output after execution in R environment of the command “citation()”. For the correct citation of the library ‘Deoptim’, the command should be look like “citation(‘Deoptim’)”. In addition, I suggest the authors to pay attention on the following article, which I will use in my following comments:

Mullen, K.M., Ardia, D., Gil, D.L., Windover, D., Cline, J. DEoptim: An R package for global optimisation by differential evolution (2011) Journal of Statistical Software, 40 (6), pp. 1-26.

2. Some details in practical implementation of algorithm shown in Chapter 4, must be clarified, despite that the most of the formulas’ notations in previous chapters are adjusted to software implementation. It is very important for experiment reproducibility. They are:

a. Mutant constant (line 386). Is the sum of all possibilities is equal to 1?

b. What is the purpose of crossover constant. How its change could affect the model and results.

c. Having the required parameters of lower and upper bounds as important input parameters in DEoptim function, their values must be cited in chapter 5. This remark is also valid for all other important parameters, such as used strategy in demonstrated experiment (see Mullen at. Al. and R manual).

3. I confirm that the orchard’s geometric form and trees distribution are very important due to 2 major reasons. Firstly, because the advantage of used optimisation method is less advantages on the regular grid than others. Secondly, due to practical implementations. Thus, the selected L- and X- shaped area are very important and even more interesting than rectangle one. However, despite that in the beginning of chapter 4, it is stated that “randomly generated set of trees in an orchard in Fig 1” is used; in the following figures (from 3 to 5) the trees are regularly distributed. What is the reason for this or they are only graphics typos?

4. Because the number of hives is discrete number, this makes the boundary range of covered area by fixed number of nesting aids very loose. Even more, there are areas of overlapping trajectories from 2 or more bee nests. Thus, the optimisation with large overlapped area could be less effective. Because of this, the only available estimate parameter of covered area seems that it is not enough. All of this implies the need of more detailed and simulated experiments in chapter 5. For instance, to demonstrate how the number of bees in hives impact on distribution and etc. The multiple results must be provided and compared by some additional statistical measure and/or comparative graphics in addition to this of covered distance.

5. Having the random initialisation, such as 'rnd <0,max>', in the beginning of every procedure, the stability confirmation by multiple replications of the very same experiment is required. How many times the obtained coordinates in demonstrated case studies are confirmed by 1000 repetitions? What about the number of the percentage of non−convergent runs (see Mullen at. Al.)?

6. PLOS authors have the option to publish the peer review history of their article (what does this mean?). If published, this will include your full peer review and any attached files.

Reviewer #1: No

Reviewer #2: No

---

## [Author Response · Author response to Decision Letter 0]

10 Dec 2020

Responses to reviewers' comments:

We would like to thank editor and reviewers for inspiring reviews and their comments and suggestions that lead to improvements to the submitted article. Reviewers' comments were included as follows:

Reviewer #1: The proposed methodology for achieving the stated objective is well described facing formal aspects and illustrating by means of a real data application. The background and references include all the works treating the issues faced in this paper. The work is well-structured starting from a description of some preliminary concepts, a review of the state of the art and then proposing an improved methodology.

Regarding introduction:

Once the introduction is read, the reader can guess the main objective, but it seems to be several the specific objectives to get the main one. The problem is that is not written clearly nor highlighted in the introduction what is this main goal. In fact, a reader like me identify the main goal at line 90, but the introduction follows another 76 lines with an important and detailed review (I like this) of the models and different solution approaches. Perhaps would be more appropriated this review if it were more concise and at the beginning of the next section, even dividing at the beginning of the two next sections.

The main idea of the presented article is incorporated into the abstract to mention it earlier. Please see lines 29-30 in MS Word document named “Revised Manuscript with Track Changes”.

In order to introduce the reader to the problematics, the introduction part seems to be extensive, both in terms of economic benefits and model approach. The authors would like to explain, motivate, and emphasize the importance of the problem analyzed at the beginning of the article.

Regarding section two:

The authors suppose randomly located trees but distances between then are known (in a compact set I guess). I do not know if the consideration of random distances would be of interest, since the first objective (line 175) is to minimize the nesting aids and. In my opinion, random locations but pre-fixed distances is not so random, surely uniformly randomized distributed. I apologize in advance if I am confused.

We have incorporated your comments into the text. Please see lines 375-377 and 427-430. 

On the other hand the formulations of the problem in equations (1), (2), (7) and (8) would be more illustrative if this equations write the minimum in what (as they do in equations (10),(11) and (13)), not only min, because this notation forces the reader going reading back.

In line 232 the authors affirm that the problem is intractable with classical optimization techniques. A short sentence explaining why would be of interest.

We have incorporated your comments into the text. Please see lines 234 - 238. 

Finally, once the original problem is rewritten by means of a metaheuristic approach, may be of interest to justify how this formulation is equivalent to the first one, i.e. the solution of the second provides the solution of the first one.

We have incorporated your comments into the text. Please see lines 273-279. 

Regarding section three:

To my knowledge, there are different metaheuristics approaches, each one with its advantages and disadvantages. Perhaps a short sentence that justify this choice would be enrichment.

We have incorporated your comments into the text. Please see lines 273-279. 

 

Reviewer #2: The authors of “Location problem of Osmia cornuta nesting aids for optimum pollination” propose a practical solution for optimisation of bees nesting aids locations. The idea of usage of differential evolution algorithm for this purpose is interesting and promising, very well explained and therefore motivated in the first three chapters of the proposed text. I think that minor revision/clarifications are required there. However, the experimental part is not very supportive and some of the claims are not clearly demonstrated in the following chapters. All my recommendations and questions are listed below:

1. At first, the citation of R and ‘Deoptim’ library is not complete. The R version must be cited as it is shown in the output after execution in R environment of the command “citation()”. For the correct citation of the library ‘Deoptim’, the command should be look like “citation(‘Deoptim’)”. In addition, I suggest the authors to pay attention on the following article, which I will use in my following comments:

Mullen, K.M., Ardia, D., Gil, D.L., Windover, D., Cline, J. DEoptim: An R package for global optimisation by differential evolution (2011) Journal of Statistical Software, 40 (6), pp. 1-26.

We have incorporated your comments into the text. Please see lines 392 in MS Word document named “Revised Manuscript with Track Changes”.

2. Some details in practical implementation of algorithm shown in Chapter 4, must be clarified, despite that the most of the formulas’ notations in previous chapters are adjusted to software implementation. It is very important for experiment reproducibility. They are:

a. Mutant constant (line 386). Is the sum of all possibilities is equal to 1?

Two different tasks have been addressed, the sum of which doesn't need to be equal to 1. In our case, the values were set based on testing performed in previous works In the submitted article, we refer the reader to source [63] and [74] in line 498-400.

Example of parameter testing:

“A disadvantage of algorithm of differential evolution, as well as of other evolutionary approaches, is that it has a dependence on the control parameter settings. Due to this fact, our effort was to determine effective settings of the parameters f and cr. The tests were done on above mentioned data with the simultaneous use of the set np = 300 a g = 500. The tested values of parameters f and cr were set as sequence of levels 0.1, 0.2, 0.3, 0.4, 0.5, 0.6, 0.7, 0.8, 0.9. The interval limits were not considered during testing (purely deterministic and purely stochastic nature of the algorithm). For each combination of pairs, ten experiments were conducted (the computing time was in range 8–11 seconds in all cases).

b. What is the purpose of crossover constant. How its change could affect the model and results.

We have incorporated your comments into the text. Please see lines 331-338. 

c. Having the required parameters of lower and upper bounds as important input parameters in DEoptim function, their values must be cited in chapter 5. This remark is also valid for all other important parameters, such as used strategy in demonstrated experiment (see Mullen at. Al. and R manual).

We have incorporated your comments into the text. Please see lines 413-416 and 446 -451

3. I confirm that the orchard’s geometric form and trees distribution are very important due to 2 major reasons. Firstly, because the advantage of used optimisation method is less advantages on the regular grid than others. Secondly, due to practical implementations. Thus, the selected L- and X- shaped area are very important and even more interesting than rectangle one. However, despite that in the beginning of chapter 4, it is stated that “randomly generated set of trees in an orchard in Fig 1” is used; in the following figures (from 3 to 5) the trees are regularly distributed. What is the reason for this or they are only graphics typos?

We have incorporated your comments into the text. Please see lines 427-430.

4. Because the number of hives is discrete number, this makes the boundary range of covered area by fixed number of nesting aids very loose. Even more, there are areas of overlapping trajectories from 2 or more bee nests. Thus, the optimization with large overlapped area could be less effective. Because of this, the only available estimate parameter of covered area seems that it is not enough. All of this implies the need of more detailed and simulated experiments in chapter 5. For instance, to demonstrate how the number of bees in hives impact on distribution and etc. The multiple results must be provided and compared by some additional statistical measure and/or comparative graphics in addition to this of covered distance.

The presented model aims primarily to minimize the number of nesting aids placed for practical reasons of maintenance. This first priority level criterion also ensures that overlaps are minimized. The second priority level criterion minimizes the total distance flown by bees. The authors have already addressed the idea of equable utilization of individual nesting aids (beehives). An additional objective function has been introduced into the problem for equable utilization. The obtained results represented a minimal deviation from the calculations made using the two objective functions model presented in the article. Proposed experiments with different numbers of nesting aids would lead to a solution to the classical location problem and would lose the original approach presented in the report.

5. Having the random initialisation, such as 'rnd <0,max>', in the beginning of every procedure, the stability confirmation by multiple replications of the very same experiment is required. How many times the obtained coordinates in demonstrated case studies are confirmed by 1000 repetitions? What about the number of the percentage of non−convergent runs (see Mullen at. Al.)?

The structure of the task has always led to convergence. In our opinion, the 100% convergence rate is because the first phase ensures a sufficient quantity of nesting aids is met. And thus, after the first phase, the number of local minimums is reduced. This way, constructed task leads to a relatively easy transition from the local minimum to the global optimum.

---

## [Editor Report · Decision Letter 1]

14 Dec 2020

Location problem of Osmia cornuta nesting aids for optimum pollination

PONE-D-20-30891R1

Dear Authors,

We’re pleased to inform you that your manuscript has been judged scientifically suitable for publication and will be formally accepted for publication once it meets all outstanding technical requirements.

Kind regards,

Dejan Dragan, PhD

Academic Editor

PLOS ONE

Additional Editor Comments (optional):

All comments were appropriately followed in the paper. Accordingly, the paper deserves an opportunity to be accepted. AE DD
---

## [Editor Report · Acceptance letter]

21 Dec 2020

PONE-D-20-30891R1 

Location problem of *Osmia cornuta* nesting aids for optimum pollination 

Dear Dr. Reiff:

I'm pleased to inform you that your manuscript has been deemed suitable for publication in PLOS ONE. Congratulations! Your manuscript is now with our production department. 

Kind regards, 

on behalf of

Dr. Dejan Dragan 

Academic Editor

PLOS ONE